# EPIC Fields
# Marrying 3D Geometry and Video Understanding

Vadim Tschernezki[★♥♦]    Ahmad Darkhalil[★♣]    Zhifan Zhu[★♣]
David Fouhey[♠]    Iro Laina[♥]    Diane Larlus[♦]    Dima Damen[♣]    Andrea Vedaldi[♥]

[♥]VGG, University of Oxford    [♣]University of Bristol
[♠]New York University    [♦]NAVER LABS Europe    [★]: Equal Contribution

## Abstract

Neural rendering is fuelling a unification of learning, 3D geometry and video understanding that has been waiting for more than two decades. Progress, however, is still hampered by a lack of suitable datasets and benchmarks. To address this gap, we introduce EPIC Fields, an augmentation of EPIC-KITCHENS with 3D camera information. Like other datasets for neural rendering, EPIC Fields removes the complex and expensive step of reconstructing cameras using photogrammetry, and allows researchers to focus on modelling problems. We illustrate the challenge of photogrammetry in egocentric videos of dynamic actions and propose innovations to address them. Compared to other neural rendering datasets, EPIC Fields is better tailored to video understanding because it is paired with labelled action segments and the recent VISOR segment annotations. To further motivate the community, we also evaluate three benchmark tasks in neural rendering and segmenting dynamic objects, with strong baselines that showcase what is not possible today. We also highlight the advantage of geometry in semi-supervised video object segmentations on the VISOR annotations. EPIC Fields reconstructs 96% of videos in EPIC-KITCHENS, registering 19M frames in 99 hours recorded in 45 kitchens, and is available from: http://epic-kitchens.github.io/epic-fields

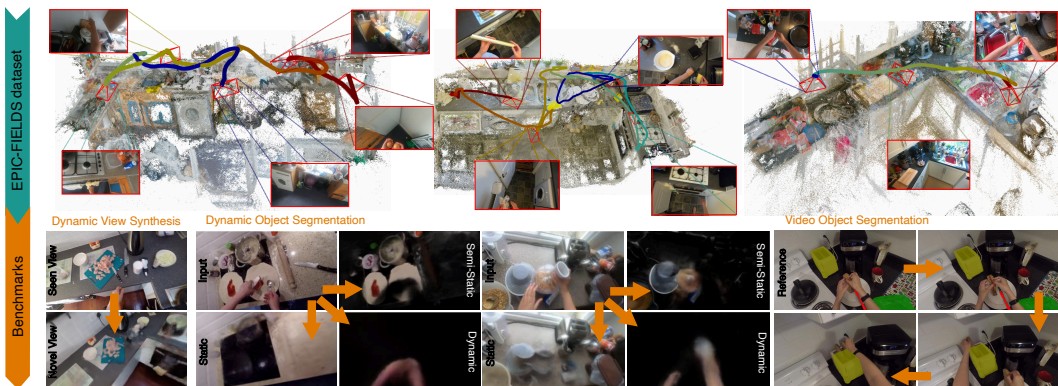

Figure 1: We propose EPIC Fields that extends EPIC-KITCHENS with 3D information, including full frame-rate camera pose trajectories (top). These are directly obtained from dynamic sequences of object interactions (sampled frames) without additional modalities or pre-scans. We showcase EPIC Fields through several benchmarks (bottom) that use the fusion of geometric and semantic cues.

37th Conference on Neural Information Processing Systems (NeurIPS 2023) Track on Datasets and Benchmarks.

# 1 Introduction

Recent breakthroughs in neural rendering [52, 33] have enabled a deeper integration of machine learning in geometric tasks like 3D reconstruction and rendering, creating a new opportunity to bring 3D geometry and video understanding closer together. By representing videos in 3D we can explain away the variability induced by the camera motion, which is dominant especially in egocentric videos. We can also integrate information extracted from each frame independently into a global, consistent interpretation of the video, as demonstrated by semantic neural rendering [74, 58, 20, 68, 21, 12, 61]. However, such successes have been mostly limited to *static scenarios*, where only the camera moves. Indeed, 3D reconstruction still struggles with dynamic content and much work remains before we can have a 3D understanding of dynamic phenomena like actions and activities.

An obstacle to further progress in 3D video understanding is the lack of suitable development data. In this paper, we address this gap by introducing *EPIC Fields*, an extension of the popular EPIC-KITCHENS [5] dataset which adds reconstructed 3D cameras and new benchmark tasks assessing both 3D reconstruction and semantic video understanding.

We choose to build on EPIC-KITCHENS because it is an established benchmark for 2D video understanding with rich annotations. Furthermore, it contains egocentric videos which are likely to benefit from 3D understanding, but which also challenge existing 3D reconstruction techniques due to their highly dynamic content and long duration (up to one hour). Dynamics include the motion of the actor and of the objects that they manipulate, as well as object transformations (*e.g.*, slicing a carrot). Furthermore, most objects are mostly static, moving only during brief spells of active manipulation. These challenges push the limits of current dynamic 3D reconstruction methods, which are usually restricted to short videos or focus on one class of objects [44, 41, 39, 26, 22, 53].

Obtaining camera information for EPIC-KITCHENS is challenging since structure-from-motion methods often fail on such complex videos. By solving this problem, we make it much easier for other researchers to start on 3D video understanding even when they are not experts in 3D vision, similar to what the setup and data introduced in works like NeRF [33] has done for static 3D reconstruction. Furthermore, while we propose specific benchmark tasks, we anticipate that researchers will be able to use our dataset to investigate many more questions than the ones we investigate here.

In summary, our first contribution is to augment EPIC-KITCHENS with camera information. To overcome the limitations [17] of traditional structure-from-motion pipelines [48], which struggle with egocentric videos, we introduce a pre-processing step that intelligently sub-samples frames from these videos, resulting in higher reconstruction reliability and speed. Our second contribution is to introduce new benchmark tasks that require or can benefit from the 3D cameras: dynamic novel view synthesis (*i.e.*, reconstructing unseen frames given a subset of frames from a monocular video); identifying and segmenting objects that move independently from the camera; and semi-supervised video object segmentation. These benchmarks use and extend the VISOR [8] annotations to provide dense ground-truth semantic labels. We report a number of baselines and conclude that, while 3D reconstruction can indeed benefit video understanding, existing approaches are challenged by the dynamic aspects of EPIC Fields.

# 2 Related work

**Egocentric action understanding using 3D.** Some egocentric datasets [38, 7, 15] contain static 3D scans of the recording locations. These typically do not contain actions, or the environments are scanned post-hoc, usually with an additional step. For instance, [38] uses stereo egocentric cameras, but no activities, and in [7, 15], reconstruction is done afterwards via hardware or additional dedicated scans. These scans are costly, which is why just 13% of Ego4D [15] data comes with a 3D scan. In contrast, we provide a pipeline for estimating camera poses from egocentric data without additional hardware or scans, which we demonstrate on an existing, challenging dataset EPIC-KITCHENS.

**Inferring cameras in egocentric videos.** In this work, we perform the challenging task of reconstructing 3D camera poses from egocentric videos that show dynamic activities from a single camera. Since the EPIC-KITCHENS [6] dataset is unscripted, the videos show natural interactions by participants in their homes, who act swiftly due to familiarity. Prior work [17, 35, 54] on these videos highlights the challenge. In [17], where ORB-SLAM was used to find short clips where the camera pose was stable, the authors note that bundle adjustment failed and reconstructions lasted for

just 7 second intervals. Using [17], [35] found hot-spots, but commented that just 44% of the frames could be registered. Others have used additional hardware information; for instance, [54] proposed using IMU data to establish short-term trajectories. In contrast, this work shows how to reconstruct cameras for *full* videos in EPIC-KITCHENS, without additional assumptions, data, or hardware.

**Multi-view videos.** A different approach to enabling neural rendering is calibrated multiview setups. Many of these datasets, however, capture humans in a "blank context", including HumanEva [51], Human3.6M [19], AIST++ [60, 24], and ZJU-Mocap [42]. There are datasets capturing humans in complex environments, such as the Immersive Light Field dataset [1], NVIDIA Dynamic Scene Datasets [70], UCSD Dynamic Scene Dataset [29], and Plenoptic Video datasets [25]. However, these videos are short (1–2 min) and, due to the capture setup, show actions outside of their natural environment. In contrast, EPIC-KITCHENS is captured with an egocentric camera and shows long captures of indoor activities. Our contribution of reconstructing the cameras over time turns the egocentric data into the multiview data needed while retaining the naturalness of the data.

**NeRF and dynamics.** NeRF extensions to dynamic data can be roughly divided into approaches that add time as an additional dimension of the radiance fields [32, 59, 66, 13, 64, 25, 47, 2] and those that instead model explicitly 3D flow and reduce the reconstruction to a canonical (static) one [44, 39, 70, 40, 63, 57, 26, 9, 72, 53, 18, 11, 27, 30]. While these methods demonstrate successes, their success depends on the dominance of camera motion over scene motion [14]. Scene motion by dynamic objects is not always common in existing datasets. Our proposed EPIC Fields contains both camera motion and fast continuous motion by the actor visible in the camera's field of view.

**NeRF and semantics.** Authors have already noted that neural rendering and 3D geometry can be helpful allies of video understanding. For instance, Semantic NeRF [74, 61] proposes to predict dense semantic labels in addition to RGB colours, while [21, 12, 50, 62] consider panoptic segmentations (things and stuff). [58, 20, 28] propose to fuse semantic features from pre-trained ViTs [3, 23, 56] into a neural reconstruction. [68, 73] represent a scene as a composition of static objects given their 2D masks. Several studies employ neural rendering to separate scenes into objects and background either without or with weak supervisory signals [10, 67, 71, 59, 49, 37, 34, 65]. With a few exceptions [59, 28, 65], however, little work has been done on decomposing *dynamic* scenes into objects.

## 3   The EPIC Fields dataset

We introduce here the new *EPIC Fields* dataset. We first describe the content of the dataset and then the process of constructing it, including several technical innovations that made it possible.

### 3.1   EPIC Fields in a nutshell

EPIC Fields extends EPIC-KITCHENS to include camera pose information. EPIC-KITCHENS contains videos of cooking activities collected using a head-mounted camera in 45 different kitchens. It has semantic annotations for fine-grained actions and their action-relevant objects, including 90K start-end times of actions [6]. VISOR [8] adds 272K manually annotated masks and 9.9M interpolated masks of hands and active objects. With EPIC Fields, we further contribute camera extrinsic parameters for each video frame as well as camera intrinsic parameters. Using the technique described in Section 3.2, we successfully processed 671 videos spanning all 45 kitchens, resulting in 18,790,333 registered video frames with estimated camera poses.

**Motivation.** Our camera annotations facilitate reconstructing and interpreting videos in 3D. Figure 2 illustrates this point by mapping some 2D action annotations from EPIC-KITCHENS to the 3D space. Lifting annotations to 3D puts them in the wider context of the environment where actions occur, and enables studying the relevance of 3D egocentric trajectories to actions (for anticipation), objects (for understanding object state changes), and hand-object understanding. The figure also illustrates mapping hand meshes extracted using [46] to the 3D context of the kitchen.

**Ethics, licensing, data protection.** EPIC-KITCHENS was collected with ethics approval by the University of Bristol and explicit consent from the participants. The data does not contain personal identifiable information or offensive content and is provided under a non-commercial license. EPIC Fields is released under the same terms.

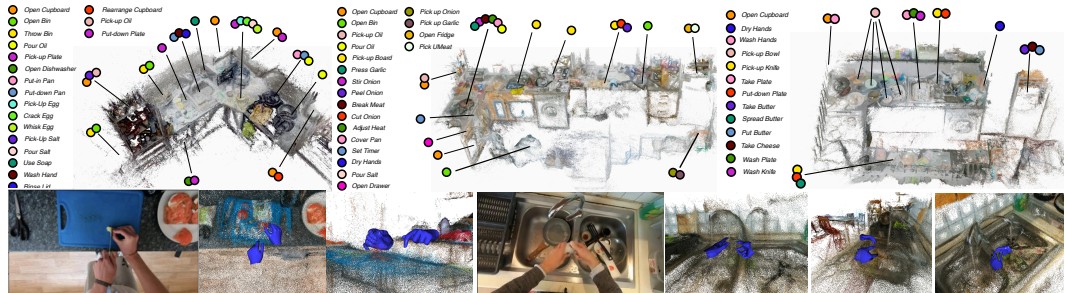

Figure 2: EPIC Fields unlocks applications that combine interactions with 3D information. We showcase examples of actions grounded in 3D (top row), and examples of integrating single-image 3D hands [46] into the kitchen reconstruction during interactions (bottom row).

## 3.2 Dataset construction

Because EPIC-KITCHENS videos were not collected with 3D reconstruction in mind they are difficult to reconstruct. For instance, they contain many dynamic objects: hands are visible in 95% of the frames and the focus of attention is often an object actively manipulated. Standard reconstruction pipelines operate under the assumption that the scene is static and are thus only moderately robust to dynamic objects. Other challenges include the video length (~9 mins on average) and the skewed distribution of viewpoints: videos alternate phases of small motion around hot-spots (*e.g.*, cooking at a hob or washing at the sink) and fast motion between hot-spots (*e.g.*, moving the pot to the sink).

We address these challenges by: (1) filtering videos to reduce the number of redundant frames, computational cost, and skew; (2) using structure from motion (SfM) to reconstruct the scene from the filtered frames; (3) registering the remaining frames to the sparse reconstruction. We accept a video's reconstruction if 70% or more of its frames are registered successfully. In this manner, we can reconstruct 96% of all EPIC-KITCHENS videos. We next describe each step, with details in the supplement.

**Frame filtering.** The goal of frame filtering is to downsample a video to reduce redundancy and skew while maintaining sufficient viewpoint coverage for accurate reconstruction. We filter frames by seeking temporal windows where frames have substantial visual overlap and then only keep one frame per window, similar to redundant frame mining [48, 55] and other SfM or SLAM pipelines. Overlap between frames is measured by estimating homographies by matching SIFT features [31]. Given a homography $H$ between two frames, we define their visual overlap $\tilde{r}$ to be the fraction of the first frame area covered by the quadrilateral formed by warping the second frame corners by $H$. Windows are formed greedily, finding runs of frames $(i + 1, \ldots, i + k)$ with overlap $\tilde{r} \geq 0.9$ to the first frame $i$ and discarding them. Filtering discards on average about 82% of frames in each video while also retaining a sufficient number of frames in the critical transitions between hot-spots.

**Sparse reconstruction.** The filtered frames are fed to an off-the-shelf structure-from-motion pipeline. Among these, we found COLMAP [48] to be more effective than VINS-MONO [45], which suffered from frequent drifts and restarts.

In Table 1 we analyse the effectiveness of the homography-based filtering algorithm by comparing it to a naïve filter that subsamples frames uniformly. We use 30 randomly selected videos for this experiment and report two standard SfM metrics [48]: the average reprojection error and the number of 3D points in the reconstruction. The first metric is a proxy for the accuracy of the reconstruction, and the second for its coverage. Both filtering techniques reduce the number of frames equally and thus result in similar computational complexity. However, homography-based filtering also addresses the skew and results in a significantly better success rate, increased coverage, and reduced reprojection error compared to uniform subsampling. Besides considering the number of points reconstructed, Figure 3 shows qualitatively the notably improved coverage obtained by homography-based filtering.

**Dense reconstruction, automated verification, and restart.** After obtaining the sparse reconstruction from the filtered subset of video frames, we use COLMAP to register the remaining frames against it, which is computationally cheap. We accept the final reconstruction if ≥70% of the video's frames, at full frame rate, are registered successfully. This process succeeds in 90% (631 videos)

Table 1: **Impact of frame filtering on the reconstruction quality.** We compare the sparse reconstruction of 30 videos using either homography-based or uniform frame filtering. Naïve uniform sampling results in only 27 of the 30 videos being reconstructed successfully (*i.e.*, dense registration rate ≥ 70%). Furthermore, the successful reconstructions have significantly reduced coverage (-16.64%) and increased reprojection error (+4.76%) compared to homography-based filtering.

| Frame sampling | Avg. #3D Points | Avg. Repr. Error | Avg. Reg. Rate | Successful Reconstructions |
|---|---|---|---|---|
| Homography-based (ours) | 27,763 | 0.798 | 98.6% | 30/30 |
| Uniformly | 23,142 | 0.836 | 89.0% | 27/30 |
| Relative change | -16.64% | 4.76% | 9.77% | -10 % |

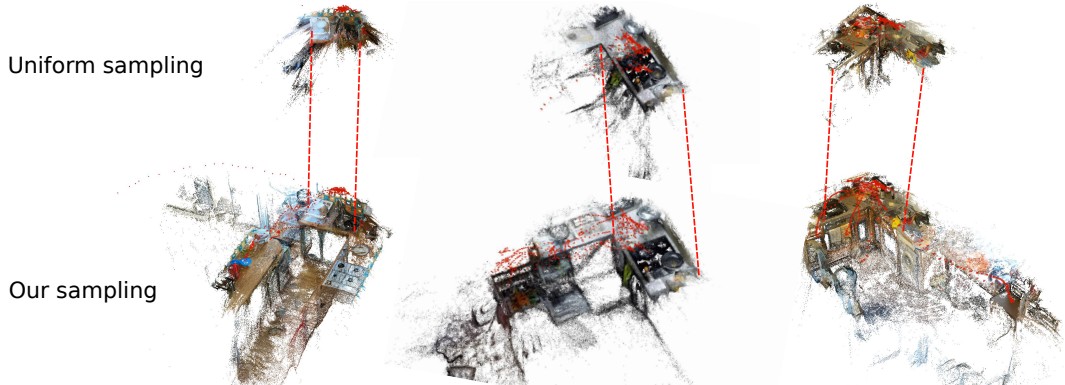

Uniform sampling

Our sampling

Figure 3: **3D reconstructions with different sampling.** We compare three scenes reconstructed using either uniform frame selection or our homography-based pipeline. Uniform sampling yields partial reconstructions with limited coverage. Ours demonstrates superior performance, resulting in better coverage by registering successfully more viewpoints.

of cases. When a video is rejected, the reconstruction process is attempted again with a higher threshold $\tilde{r} \geq 0.95$; this usually doubles the number of frames that COLMAP needs to process for the reconstruction, but increases the success rate to 96%. We discuss reasons for the failure of the last 29 EPIC-KITCHENS videos in the supplement.

**Application to other egocentric videos.** While we developed our reconstruction pipeline by considering the EPIC-KITCHENS data, the approach we obtained is general and applies equally well to other egocentric video collections such as Ego4D [16], at least for indoor locations. We give examples of these reconstructions in the supplement.

## 4 The EPIC Fields benchmarks, experiments and results

We define three benchmarks on EPIC Fields that probe 3D video understanding. Annotations, evaluation code and baselines are released as part of EPIC Fields; further details are in the supplement.

### 4.1 Dynamic New-View Synthesis (D-NVS)

Given a subset of video frames as input, the goal of dynamic new-view synthesis (D-NVS) is to predict other video frames given only their timestamps and camera parameters. While other D-NVS benchmarks exist, EPIC Fields is more challenging due to the first-person perspective and the large number of dynamic objects. In Table 2 we compare EPIC Fields to commonly used datasets in D-NVS. EPIC Fields offers a significant step up in complexity and scale with *significantly longer* videos and associated semantics. For detailed statistics, please refer to the supplement.

**Video selection.** Due to the computational cost of most D-NVS algorithms, we limit the D-NVS benchmark to a subset of 50 videos (14.7 hours and 2.86M registered frames) extracted from the train/val set of VISOR [8] (this selection includes 96.1% of the frames annotated in VISOR).

Table 2: Comparison of datasets commonly used in dynamic new-view synthesis.

| Dataset | #Scenes | Seq. Length | Monocular | Semantics |
|---------|---------|-------------|-----------|-----------|
| Nerfies [40] | 4 | 8–15 sec | ✗ | ✗ |
| D-NeRF [44] | 8 | 1–3 sec | ✗ | ✗ |
| Plenoptic Video [25] | 6 | 10–60 sec | ✗ | ✗ |
| NVIDIA Dynamic Scene Dataset [70] | 12 | 1–5 sec | 4 / 12 | ✗ |
| HyperNeRF [41] | 16 | 8–15 sec | 13 / 16 | ✗ |
| iPhone [14] | 14 | 8–15 sec | 7 / 14 | ✗ |
| SAFF [28] | 8 | 1–5sec | ✗ | ✓ |
| **EPIC Fields [D-NVS]** (ours) | 50 | 6–37 min (Avg 22) | 50 / 50 | ✓ |

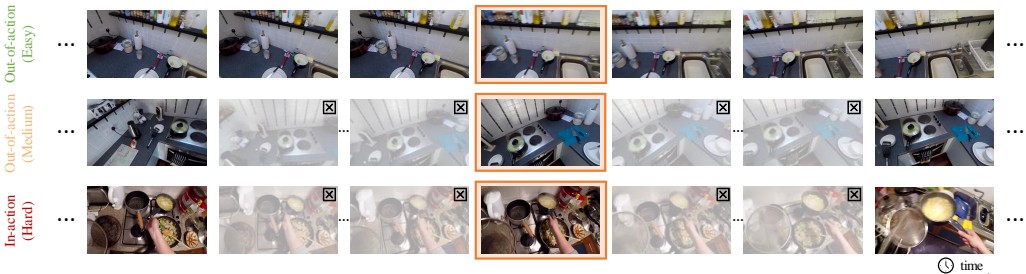

Figure 4: **Definition of the three difficulty levels for the task of dynamic new-view synthesis.** Validation and test frames are selected to meet three reconstruction difficulty levels. **In-Action frames (Hard)** happen during an action and are harder to reconstruct due to the dynamics. **Out-of-Action (Medium) frames** happen outside an action, but are far from a train frame. **Out-of-Action (Easy) frames** are near train frames. Frames in a bounding box (orange) represent either val/test frames. Frames marked with a cross are discarded to create a larger time gap around each val/test frame (medium and hard levels). All other frames can be used for training.

**Frame selection.** For each video in the D-NVS benchmark, we select the video frames to be used as input to the system (training) and those that remain unseen and are used for evaluation only (validation/testing). Specifically, we propose categorising evaluation frames into three tiers of difficulty (easy, medium, hard — visualised in Figure 4), determined by the type of motion and the temporal gap between the evaluation and training frames. **In-Action** frames correspond to common 'put', 'take', and 'cut' actions annotated in EPIC-KITCHENS, based on their start-stop times; they are characterised by substantial object motion due to hand-object interactions and are thus more difficult to reconstruct. In pursuit of a greater challenge, for the **In-Action (Hard)** set of frames, we exclude frames from the training set occurring within 1 second of a test frame. **Out-of-Action** frames occur outside action segments, where there is no appreciable motion except for the camera, making these frames generally easier to reconstruct. For the **Out-of-Action (Medium)** set, we sample 70% of the out-of-action frames with the same time gap as above. The **Out-of-Action (Easy)** set corresponds to the remaining 30% without removing the neighbouring training frames. The reasoning is that it is generally easier to predict a frame temporally close to a training one. We assign every other evaluation frame to the validation and test sets, respectively. The average time gap between consecutive evaluation frames is 3.73 seconds. Further statistics are provided in the supplement.

**Benchmark methods.** To demonstrate how EPIC Fields can be used and to probe the limits of the state of the art in such challenging scenarios, we consider three neural rendering approaches: NeRF-W [32], NeuralDiff [59], and T-NeRF+, an extended version of T-NeRF [14].

*NeuralDiff* [59] is a method tailored to egocentric videos. It uses three parallel streams to separate the scene into the actor, the transient objects (that move at some point in the video), and the background that remains static. We combine the predictions of the actor and transient objects to predict our *dynamic* and *semi-static* objects, which will be relevant in Section 4.2.

*NeRF-W* [32] augments NeRF with the ability to 'explain' photometric and environmental (non-constant) variations by learning a low-dimensional latent space that can modulate scene appearance and geometry. As a result, NeRF-W also separates static and transient components. We follow the modification from [59] to render NeRF-W applicable to video frames and the D-NVS task.

Table 3: **Dynamic new-view synthesis**. We compare different neural rendering approaches for frames from different difficulty levels (easy, medium, hard). We report PSNR considering all pixels in each test frame. Given the mask annotations from VISOR for *In-Action* frames, we also report PSNR on background (BG) and foreground (FG) pixels separately for the hard (*In-Action*) setting.

| Method | Easy | Medium | Hard | | |
| --- | --- | --- | --- | --- | --- |
| | | | All | BG | FG |
| NeRF-W [32] | 21.13 | 19.3 | 17.93 | 18.99 | 13.54 |
| T-NeRF+ [14] | 21.58 | 19.81 | 18.44 | 19.73 | 13.74 |
| NeuralDiff [59] | 22.14 | 19.88 | 18.36 | 19.54 | 13.37 |

*T-NeRF+* [14] was proposed as a baseline to evaluate state-of-the-art NeRFs on dynamic scenes. It was shown to outperform other methods in terms of the quality of the synthesised images. We extend T-NeRF by adding another stream to the time-conditioned NeRF architecture that models the background (static parts of the scene).

**Results.** To measure performance on this task, we report the Peak Signal-to-Noise Ratio (PSNR) of the test frame reconstructions, which is a proxy for the quality of the underlying 3D reconstructions with the key advantage of not requiring 3D ground-truth for evaluation. We report results in Table 3 for the three levels of difficulty. There is a strong relationship between PSNR and difficulty: PSNR is consistently lower for all methods when rendering views during actions (hard) compared to outside actions (medium, easy). Some limitations of rendering these hard test frames are shown in Figure 5. For example, the bottom row shows that no 3D baseline renders the person's arm correctly, since all models struggle to interpolate the person's movement between frames. We further observe a significant gap in rendering quality if we calculate PSNR separately for foreground and background regions. We use the VISOR annotations of hands and active objects for In-Action frames to obtain this separation. These results not only highlight the existing limitations of current methods but also offer a valuable benchmark for assessing potential improvements in a targeted manner.

### 4.2 Unsupervised Dynamic Object Segmentation (UDOS)

The goal of Unsupervised Dynamic Object Segmentation (UDOS) is to identify which regions in each frame correspond to dynamic objects. This task can be approached in 2D only but is a good proxy to assess 3D methods as well, and can, in fact, be boosted by 3D modelling. Here, we extend the setting introduced in [59], using 20× more data and adopting a more nuanced evaluation protocol.

**Video and frame selection.** We use the same selection of videos as for the D-NVS task, but only use the In-Action frames with VISOR annotations, as they provide ground-truth dynamic object segmentations. We convert VISOR masks into a foreground-background mask for each frame in three ways, depending on objects that are currently moving, or those that have moved at a different time in the video. In the **dynamic objects only** setting, the foreground contains hands and other visible body parts as well as object masks only for objects that are currently being moved. We use the VISOR contact annotations to identify these objects and augment these with additional manual masks for visible body parts including torsos, legs, and feet. More details are in the supplement. In the **semi-static only** setting, we consider only objects that moved at some point during the video, but not during the current frame. We select these objects by watching the video and identifying all objects that have moved at least once. VISOR contains annotations of these objects only on frames where they are considered *active*. We employ an automated method to propagate the annotations to cover all frames, resulting in a set of semi-static object masks. This is the complete set of masks for all objects that have moved at any point in the video, even if they are temporarily static. More details can be found in the supplement. We combine both to report the **dynamic and semi-static** setting.

**Benchmark methods.** We use NeuralDiff and NeRF-W from the NVS task, since, by design, they decompose scenes into static and dynamic components. Additional considerations are necessary to make *T-NeRF+* applicable to UDOS. In order to disentangle the modelling of both radiance fields in terms of temporal variation, we apply the uncertainty modelling from [32] to model a change in observed colours of pixels that occur due to dynamic effects inside the scene. This extension enables *T-NeRF+* to learn a decomposed radiance field.

We also consider a 2D baseline, *Motion Grouping (MG)* [69], a state-of-the-art method for self-supervised video object segmentation. It trains a segmentation model using an autoencoder-like

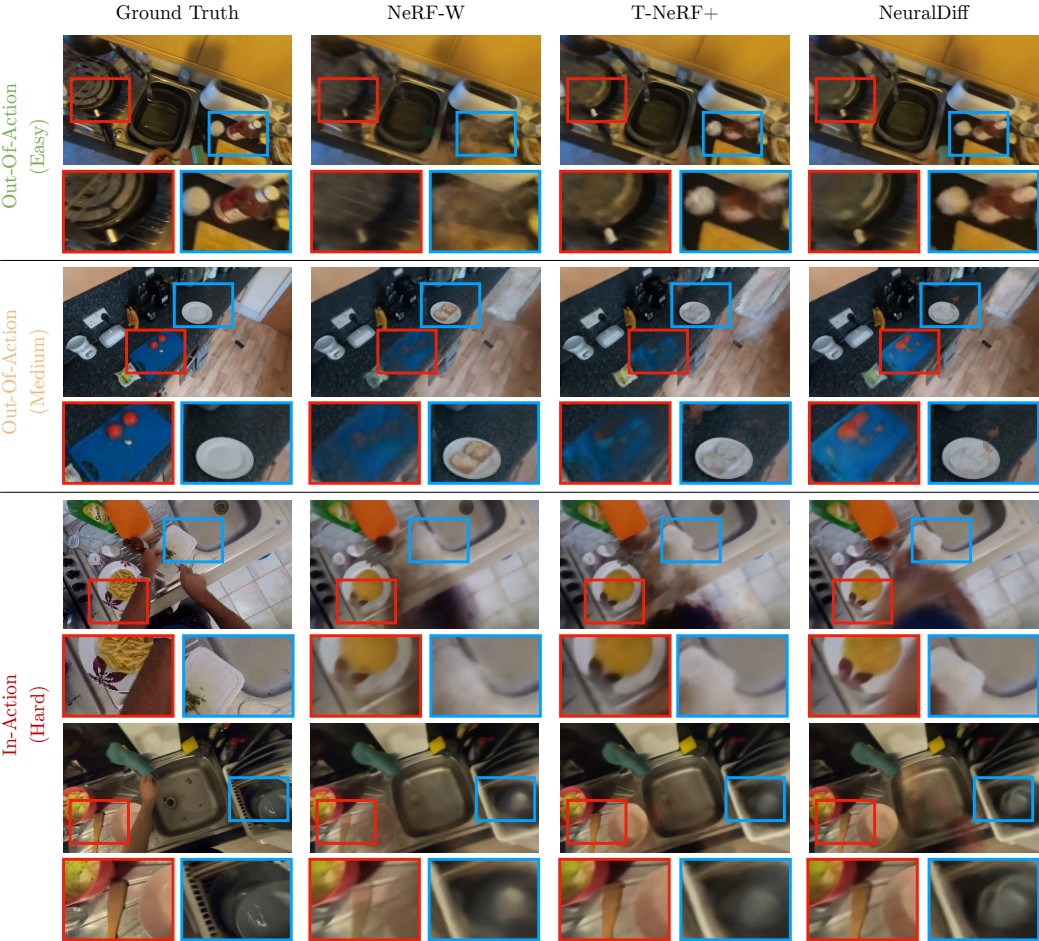

Figure 5: **Dynamic new-view synthesis.** We compare the outputs of 3D methods NeRF-W [32], T-NeRF+ [14], and NeuralDiff [58], for novel viewpoints, across three different complexity levels. The predictions are more accurate with less difficult motion as shown in the first and second row. The task becomes more challenging for our hard samples.

framework. The model has two output layers; one layer represents the background, and the other layer identifies one or more moving objects in the foreground, including their opacity layers. These layers are then linearly composed and optimised to reconstruct the input image. Since this approach is unsupervised, it can be compared fairly to the 3D baselines for this task.

**Results.** To evaluate performance, we measure 2D segmentation accuracy on test frames using mean average precision (mAP) as in [59]. Table 4 compares unsupervised 2D baselines and 3D baselines. Depending on the type of observed motion, 3D-based methods offer advantages over 2D methods and vice versa. For example, 3D-based methods are better suited for discovering semi-static objects that are not currently in motion, *i.e.*, they have been moved at different times within a video. This is evident by the improved segmentation performance when considering this type of motion (*i.e.*, *SS+D* and purely SS). However, we note that none of the 3D-based methods explicitly consider motion. Consequently, MG, which takes as input optical flow, performs better on purely dynamic motion, but struggles to segment objects that are temporarily not moving. This drawback of 3D-based methods, compared to 2D motion-based methods, underscores the current challenge in capturing dynamics in neural rendering. Addressing this limitation is an open question for future research.

Figure 6 shows qualitative results. We observe that MG performs particularly well on objects that are constantly in motion, for example, the moving body parts of the person. Among the 3D methods, NeuralDiff is better at capturing dynamic objects, and, unlike MG, both NeuralDiff and T-NeRF+ are able to segment various semi-static objects as well since they do not rely on continuous motion.

Table 4: **Unsupervised dynamic object segmentation**. We report the mean average precision (mAP) on segmenting the semi-static (SS) and dynamic components of the scene, and also their union (SS+Dyn). All methods are trained without explicit supervision, *i.e.*, no masks are used during training, only for evaluation.

| Method | 3D | SS+Dyn | SS | Dynamic |
|---|---|---|---|---|
| MG [69] | ✗ | 55.53 | 12.78 | 64.27 |
| NeRF-W [32] | ✓ | 45.62 | 20.97 | 28.52 |
| T-NeRF+ [14] | ✓ | 64.91 | 24.48 | 44.27 |
| NeuralDiff [59] | ✓ | 69.74 | 25.55 | 55.58 |

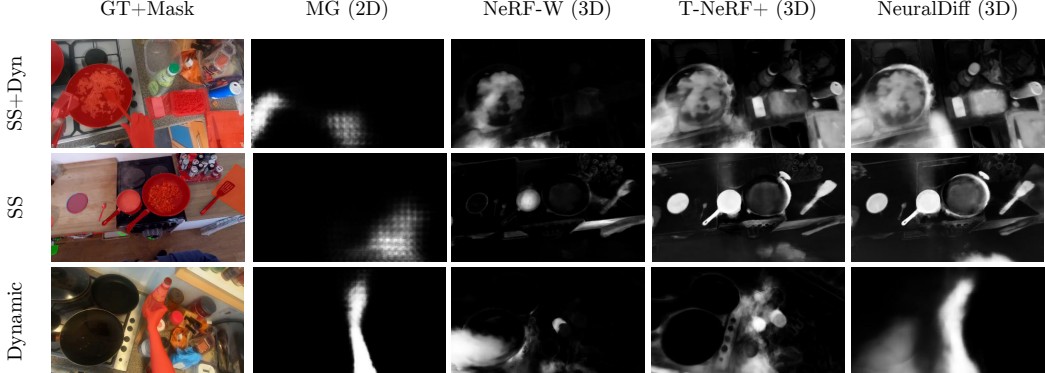

Figure 6: **Unsupervised dynamic object segmentation.** We compare three 3D baselines (NeRF-W [32], T-NeRF+ [14] and NeuralDiff [58]) and one 2D baseline (MG [69]) with three motion types. The 2D baseline captures the person and does well on the purely dynamic (short-range motion) setting. 3D models do this and also segment semi-static (SS) components (long-range motion, i.e., objects that were moved some time ago). In *SS+Dyn*, the evaluation includes SS and dynamic components.

## 4.3 Semi-Supervised Video Object Segmentation (VOS)

Semi-Supervised Video Object Segmentation (VOS) is a standard semi-supervised video understanding task: given the mask for one or more objects in a reference frame, the goal is to propagate the segments to subsequent frames. For this task, we use the train/val splits published as part of the VISOR VOS benchmark (See [8] Sec. 5.1). VOS is usually approached by using 2D models. Here, we explore how the 3D information in EPIC Fields can be used for it instead.

**Benchmark methods.** We evaluate two naïve baselines for this task, one in 2D and another in 3D. For completeness, we also compare these to existing, trained 2D VOS models.

*Fixed in 2D.* We make the assumption that the pixels in the first frame remain constant throughout the entire sequence. This naïve baseline is prone to failure when the camera undergoes movement.

*Fixed in 3D.* To better understand the potential of 3D information for VOS, we compare the 2D baseline above to a 3D one. In the 3D baseline, an object mask is projected to 3D and its position in 3D is fixed throughout the sequence. The mask is then re-projected to other frames using the available camera information. This works well for static objects and achieves two effects. First, objects can be reliably tracked over occlusions. Second, detecting when these objects are in or out of view is a by-product of estimated camera poses.

*Trained 2D models.* We also evaluate two state-of-the-art models for video object segmentation, STM [36] and XMEM [4]. These are trained on the train set of VISOR.

**Results.** We compare the baselines on the VISOR benchmark using the evaluation metrics defined in [43] which are the region similarity $\mathcal{J}$ and contour accuracy $\mathcal{F}$. We also distinguish the set of objects that are static, such as 'fridge', 'floor', and 'sink', and report the above metrics separately for these and all other movable objects (SS+Dyn). Table 5 shows the results where the *Fixed in 3D* clearly outperforms the *Fixed in 2D* by a significant margin for the anticipated *static* objects

Table 5: **Semi-Supervised VOS**. We compare naive baselines in 2D and 3D, as well as pretrained/fine-tuned models on static and dynamic objects on the validation set of VISOR VOS. *: two videos from the validation set are excluded as they don't have successful reconstructions.

| Method | 3D | VISOR VAL[8] | | | Static | | | SS + Dyn | | |
|---|---|---|---|---|---|---|---|---|---|---|
| | | $\mathcal{J}\&\mathcal{F}$ | $\mathcal{J}$ | $\mathcal{F}$ | $\mathcal{J}\&\mathcal{F}$ | $\mathcal{J}$ | $\mathcal{F}$ | $\mathcal{J}\&\mathcal{F}$ | $\mathcal{J}$ | $\mathcal{F}$ |
| Fixed in 2D | ✗ | 12.5 | 13.4 | 11.6 | 17.8 | 23.8 | 11.6 | 12.0 | 11.9 | 12.0 |
| Fixed in 3D * | ✓ | 31.3 | 30.5 | 32.2 | 48.4 | 52.2 | 44.6 | 29.6 | 27.8 | 31.5 |
| Pretrained STM | ✗ | 63.0 | 60.8 | 65.2 | 64.3 | 65.4 | 63.1 | 63.7 | 60.8 | 65.5 |
| Fine-tuned STM | ✗ | 76.4 | 74.2 | 78.6 | 76.8 | 77.7 | 76.0 | 76.6 | 73.8 | 79.5 |
| Pretrained XMEM | ✗ | 64.0 | 61.5 | 66.4 | 63.2 | 64.0 | 62.5 | 64.1 | 61.1 | 67.1 |
| Fine-tuned XMEM | ✗ | 77.3 | 75.2 | 79.4 | 77.0 | 77.7 | 77.4 | 78.0 | 75.3 | 80.7 |

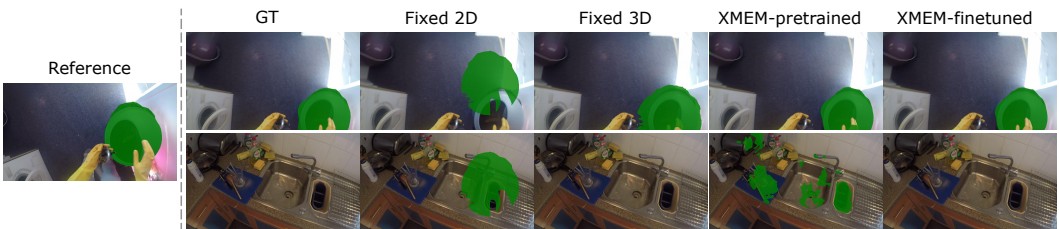

Figure 7: **Semi-Supervised Video Object Segmentation.** We compare our baselines on two frames from the same sequence. The *Fixed in 3D* baseline can track the bin over camera motion and recognise in/out-of view. Pretrained models usually suffer from false positives in the out-of-view scenes.

(+30.6%) but also improves results for the remaining *semi-static and dynamic* (+17.6%) objects. This is because such objects do remain unmoved for some duration of the videos. This highlights the additional value derived from representing objects in 3D. Figure 7 visualises one example where the bin is successfully propagated using the *Fixed in 3D* baseline, including when out of view. The pretrained models struggle to propagate masks for the novel objects in the dataset or for masks that go out of the scene. These are cases that the *Fixed in 3D* baseline successfully handles. However, the fine-tuned models are quantitatively and qualitatively superior as they are trained on the dataset. No prior work has utilised 3D information along with learnt models for the task of semi-supervised VOS. We hope our novel benchmark can trigger new VOS approaches that tackle the combined challenge of keeping track of static objects in 3D and dynamic objects through trained propagation of objects during motion and transformations.

## 5 Conclusions

We introduced EPIC Fields, a dataset to study 3D video understanding. We addressed the difficult challenge of reconstructing cameras in EPIC-KITCHENS videos, introducing filtering and other techniques that are portable to other similar reconstruction scenarios. Using these pre-computed cameras facilitates working on 3D video understanding even without significant expertise in photogrammetry.

With EPIC Fields we also defined three benchmark tasks: dynamic new-view synthesis, unsupervised dynamic object segmentation, and video object segmentation. Our results show that the performance of state-of-the-art dynamic neural reconstruction/rendering methods strongly depends on the type of motion. In particular, the gap in reconstruction quality between the dynamic and the static parts of the videos show that there is ample margin for further improvements in the handling of dynamic objects. Similar findings apply to the segmentation of objects, where 3D-based models can assist unsupervised video object segmentation and propagate masks of static objects over time. We hope that these results, the proposed benchmark data and code (comprising evaluation, camera reconstruction, and baselines) will assist the community in investigating further methods that combine geometry and video understanding.

**Societal impact.** While we expect that our benchmark will lead to positive impact, including applications to augmented and mixed reality including AR assistants, there are potential negative impacts as well: better AR may be used for deception and many capabilities powering an assistant may also aid surveillance.

**Acknowledgements** Project supported by EPSRC Program Grant Visual AI EP/T028572/1. A. Darkhalil is supported by EPSRC DTP program. Z. Zhu is supported by UoB-CSC Scholarship. I. Laina and A. Vedaldi are supported by ERC-CoG UNION 101001212. D. Damen is supported by EPSRC Fellowship UMPIRE EP/T004991/1. D. Fouhey is supported by a NSF CAREER #2142529.

We thank Sanja Fidler and Amlan Kar, from the University of Toronto, for contributions to the initial idea and discussions of this project. We also thank Rhodri Guerrier for assisting in manual annotations of the test set.

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
