# EPIC Fields
# Marrying 3D Geometry and Video Understanding
# Supplementary Material

**Vadim Tschernezki**[★♥♣]    **Ahmad Darkhalil**[★♣]    **Zhifan Zhu**[★♣]
**David Fouhey**[♠]    **Iro Laina**[♥]    **Diane Larlus**[♦]    **Dima Damen**[♣]    **Andrea Vedaldi**[♥]

[♥]VGG, University of Oxford    [♣]University of Bristol
[♠]New York University    [♦]NAVER LABS Europe    [★]: Equal Contribution

In this supplementary material, we first describe the companion video that provides an overview of our dataset (Section 1) and then detail how the data was released (Section 2) along with taking stock of additional information specifically promised in the checklist (Section 3). Next, we provide additional details on the dataset construction (Section 4) and on the benchmarks (Section 5). We devote a final section (Section 6) to showing that the EPIC Fields pipeline could be applied to reconstructing videos from the Ego4D dataset.

## 1    Supplementary video

We provide a short video in the form of a trailer at `https://youtu.be/RcacE26eObE`. It allows to visually assess how challenging the reconstruction problem is and hints at how frame filtering helps. The video also illustrates how the new camera poses complement the existing semantic annotations for this dataset (hands and active objects), showcasing the potential of marrying 3D geometry and video understanding. Additionally, we provide a couple of qualitative results for static and dynamic novel view synthesis, one of the benchmark tasks we describe in the paper.

## 2    Released data

Our dataset is now publicly available with visualisation scripts that enable exploring all the reconstructions and camera poses.

The data can be downloaded from `http://epic-kitchens.github.io/epic-fields`. We released the camera parameters along with sparse point clouds (light-weight version of 10–20MB/video) as well as the full COLMAP database of dense registrations (heavy-weight version). The latter enables comparisons with the dense registrations in EPIC Fields, and also allows the use of the COLMAP library and interface for visualisation and exploration.

The webpage also includes links to the visualisation code and to the code to replicate training, inference and evaluation for our benchmarks.

## 3    Dataset and benchmark details mentioned in the checklist

**Data splits.** We provide information about the data splits used in the benchmark in Section 4.3.

**Annotations.** We offer two additional sets of manual annotations, on top of those available in VISOR [4] to facilitate the assessment of the MG, NeuralDiff, T-NeRF+, and NeRF-W benchmarks. We employ these annotations as ground truth for evaluation on the UDOS task; nevertheless, we anticipate that they may prove valuable for various applications in future research endeavors.

37th Conference on Neural Information Processing Systems (NeurIPS 2023) Track on Datasets and Benchmarks.

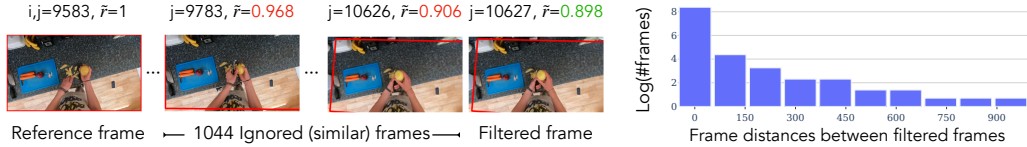

Figure 1: **Filtering frames before reconstruction.** We apply a 2D frame filtering technique to mitigate the oversampling of highly overlapping views (viewpoint distribution skews mentioned in Section 3.2 of the main paper) and to reduce the complexity of the SfM reconstruction. (Left) For a reference frame $i$, we show two of the ignored frames, the next frame after filtering, and their respective overlap $r$ score with the original frame. Filtering discards 1044 frames (ca. 17 seconds) in this case. (Right) Histogram of the distances between frames after filtering (for one video).

The first set of annotations serves the dynamic objects. We provide human body annotations for all evaluation frames, as VISOR exclusively annotates the hands but not the other visible parts of the body. To achieve this, we identify up to 3 frames per video with visible body parts of the camera wearer. Using manual points, we employ SAM [7] to generate a total of 143 automated human-body annotations. These frames serve as reference frames for the DeAOT [12] model pre-trained on YT-VOS to propagate the masks across all evaluation frames.

The second set of annotations is dedicated to semi-static objects. VISOR primarily addresses active objects within specific segments of the video, whereas our method aims to evaluate semi-static objects that may have moved at any point during the video. To achieve this, we utilize a fine-tuned MS-DeAOT [12] on VISOR along with a maximum of 10 VISOR ground truth annotations as reference frames to extend the coverage of semi-static objects across all evaluation frames. As a result, all objects that have moved during the video are annotated by a mask, on every evaluation frame.

**Hyperparameters.** We provide information about the baselines used in our benchmark and their hyperparameters in Section 5.

**Total compute used.** Estimating the precise computational budget of a multi-institution project of this scope is challenging. However, we report the actual computational time specifying the machine used in each case. All resources used were local. The main components of this project were:

- *Reconstruction:* As described in Section 4.2, the reconstruction corresponds to a total of 2264 hours of compute, 1695 hours for the sparse reconstructions and 569 for registration. This was parallelised across two machines with two GPUs each (two 11GB NVIDIA GeForce RTX 2080 Ti for the first machine, 12GB NVIDIA TITAN X and 11GB NVIDIA GeForce GTX 1080 Ti for the second machine).
- *NVS, UDOS Benchmarks:* We estimate that running the 3D baselines for D-NVS and UDOS benchmarks required 2400 GPU hours. Experiments on the D-NVS benchmark were carried out using several NVIDIA GPUs on a cluster, including P40, M40, V100, RTX8k and RTX6k. The training required up to 10GB of GPU memory for each experiment. The models for both benchmarks required a total of about 2400 GPU hours. We ran the experiments in parallel on 24 GPUs, resulting in a training time of 4.17 days. Both D-NVS and UDOS required each 50% of the total computation.
- *MG (UDOS Benchmark):* We ran this baseline on a single 16GB V100 GPU. The total training time is about 5.5 days.
- *VOS Benchmark:* The *Fixed in 3D* baseline requires next-to-no compute — homography fitting on SIFT features is calculated during the reconstruction step. However, training STM and XMEM took 1.2 and 1.4 days respectively on a single 16GB V100 GPU.

We expect that by providing both sparse and dense reconstructions to the whole community, this effort will greatly reduce computation time for all the dataset users.

## 4 Additional details on the dataset construction

### 4.1 Frame filtering

As discussed in Section 3.2 in the main paper, we downsampled videos to reduce the viewpoint skew that is common for ego-centric videos. The filtering discards on average 81.8% of all frames and allows the SfM pipeline to focus on more diverse views. Figure 1 visualises the filtering process

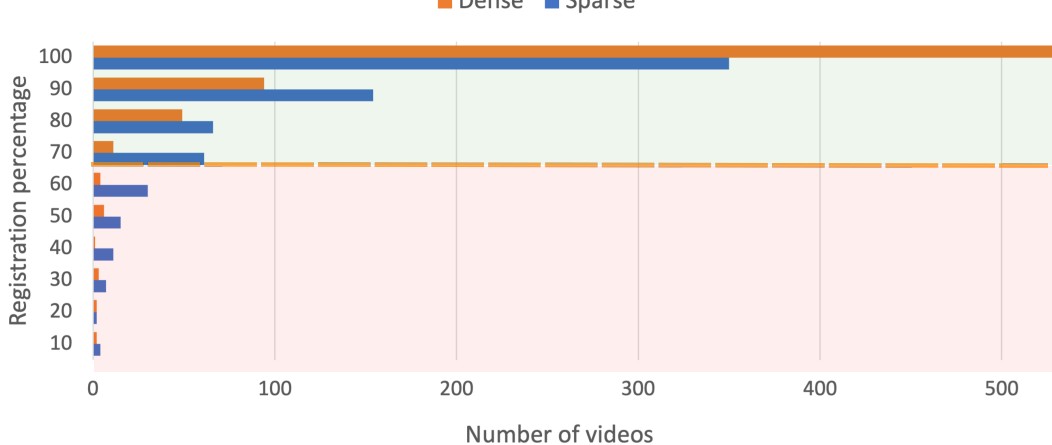

Figure 2: **Percentages of registered frames.** The dashed line specifies the threshold of the minimum dense registration rate to accept the reconstruction, otherwise, it would be considered a failure.

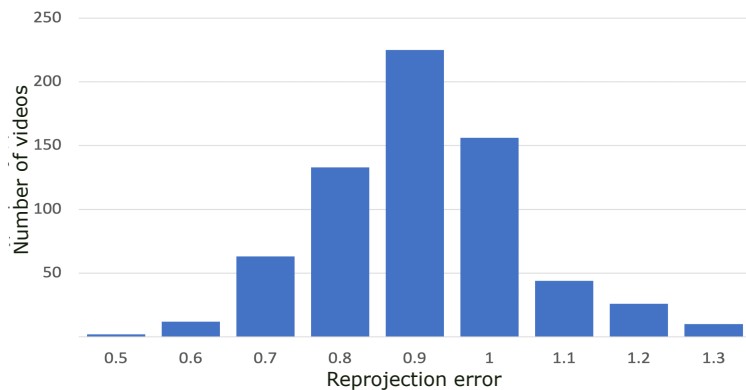

Figure 3: **Average reprojection error of EPIC Fields.** The majority of our reconstructions have an average reprojection error lower than 1.

using an example. The shown frame range contains many views that are similar to each other. The filtering discards 1044 redundant frames between frames $j = 9583$ and $j = 10627$. The figure also shows a histogram of distances between filtered frames.

### 4.2 Dataset statistics

**How do we accept/reject a reconstruction?** After producing the sparse reconstructions, we register all the frames; we then consider the videos with at least 70% dense registration rate. The histogram for both sparse and dense reconstructions is depicted in Figure 2. The majority of our reconstructions exhibit a dense registration rate exceeding 80%. In total, we successfully reconstructed 671 out of the 700 EPIC-KITCHENS videos, with average registration rates of 84.1% and 92.0% for the sparse and the dense reconstructions respectively. This is because we specifically select frames during transitions between kitchen hotspots for accurate reconstruction. This explains the higher registration rate for dense reconstructions.

**Metrics for reconstruction quality.** We use the common SfM metrics to assess the quality of the reconstructions. Figure 3 shows the histogram of the reprojection error of all the reconstructions. The average and maximum reprojection errors are 0.87px and 1.3px respectively. We use an image resolution of 456×256 to obtain the reconstructions and to calculate the reprojection errors.

**How long does the reconstruction pipeline take?** In Figure 4, for different video durations, we report the time required for the sparse reconstruction, for registration to obtain the dense reconstruction, and the total reconstruction time. As the length of the videos increases, the sparse reconstruction time

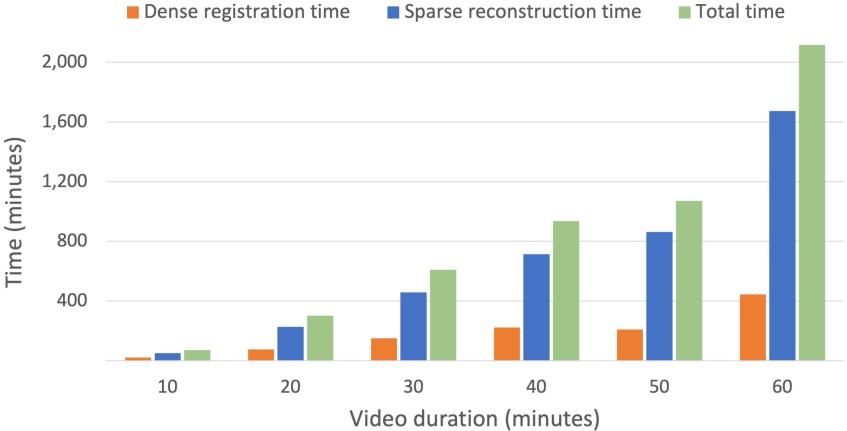

Figure 4: **Reconstruction time per video length.** We plot time for the sparse reconstruction (blue), registration time to obtain the dense camera poses (orange) and total reconstruction time (green) for different video durations. While the time for registration is almost linear, the reconstruction time increases non-linearly as a function of the video length, mainly because of bundle adjustment.

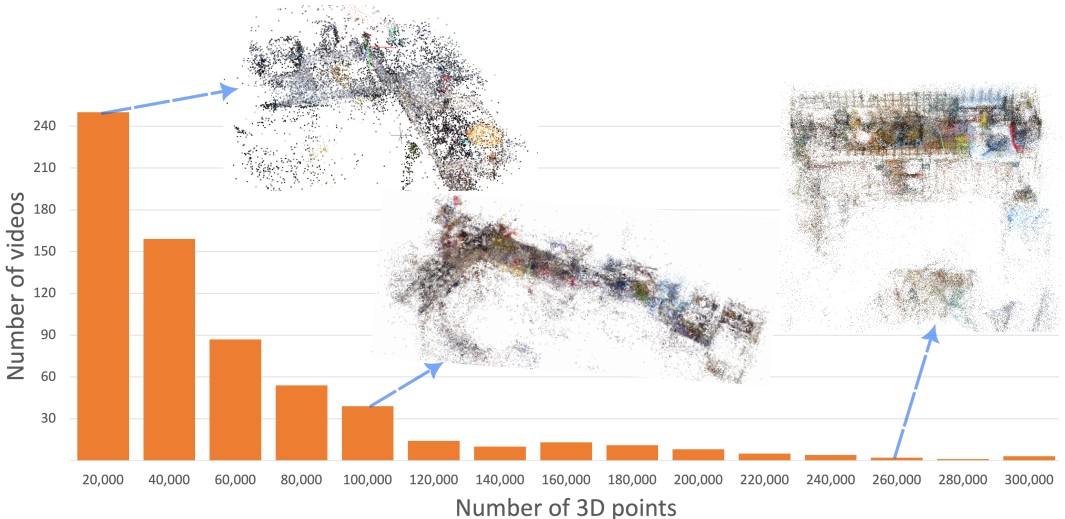

Figure 5: **Number of 3D points histogram.** The majority of our reconstructions generate fewer than 40,000 points that are enough to represent the kitchen. However, some reconstructions have more than 100,000 points, we include the point clouds for each points range showing the fine details covered by having more points.

follows a non-linear growth pattern. Overall, the sparse reconstruction and the registration processes took 1695 and 569 computation hours, respectively. We parallelise the pipeline on 2 machines with 2 GPUs each.

**How large are these reconstructions?** Figure 5 displays a histogram representing the number of 3D points in the sparse reconstructions, along with three example point clouds derived from reconstructions with varying numbers of points. These demonstrate the complexity of our reconstructions, which are capable of covering entire kitchens with fine-grained details. On average, each reconstruction consists of around 45,000 3D points.

**Reasons for the reconstruction failures.** While our reconstruction failure rate is only 4%, we examined the primary causes of these failures. These are mainly attributed to very short videos with large scene coverage, and challenging lighting conditions. (1) In the case of very short videos with large scene coverage, *e.g.*, a person just walking through the kitchen to retrieve one item and then walking out again, COLMAP often encounters difficulties due to the insufficient quantity of features

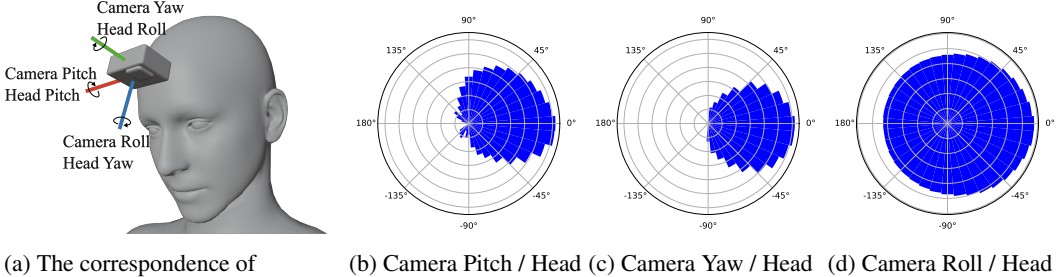

(a) The correspondence of Yaw/Pitch/Roll between the camera and the head.

(b) Camera Pitch / Head Pitch / Head Raising Up-or-Down

(c) Camera Yaw / Head Roll / Head Tilting Left-or-Right

(d) Camera Roll / Head Yaw / Neck Turning

Figure 6: Camera mounting arrangement (a) and the log-scale polar histogram of the three camera orientation parameters in the dataset (b-d).

and viewpoints. The median duration for the unsuccessful reconstructions is 1.5 minutes, compared to 6 minutes for the successful ones. This problem is exacerbated when the brief video captures a multitude of different locations within the kitchen, switching rapidly between these. (2) A couple of failure cases were linked to videos recorded under very low lighting, which led to a poor quality set of features to match. The average number of observed features per image for these unsuccessful videos was 198, compared to an average of 358 features per image for successful reconstructions.

**Distribution of camera orientations.** Figure 6 displays histograms representing the distribution of *relative* camera orientations of all EPIC Fields frames. Each frame uses the mean camera orientation within the video as reference. The histograms reveal that EPIC Fields contains diverse camera motions that are a result of natural head motions, such as looking up/down or tilting left/right. It is important to note the distinction between the camera orientations due to the particular camera mounting in EPIC-KITCHENS, illustrated in the figure. We thus particularly note camera motions and how they correspond to head motion given the specified mounting.

In summary, the figure shows larger head motion looking up (compared to the average camera orientation) than looking down, a balanced tilting as well as full 360 coverage of the kitchen by the body and/or head rotating in the scene.

## 4.3 Statistics of the benchmark splits

We provide statistics of the splits for the D-NVS task of our benchmark in Table 1. In UDOS, the objective is to segment dynamic and semi-static objects in videos without relying on supervision from ground-truth segmentations during training. Thus, *all* frames are observed during training. Evaluation frames are the same as for the D-NVS task. For the VOS task, we use the train/val splits published as part of the VISOR VOS benchmark (See [4] Sec. 5.1).

For D-NVS, we divide the evaluation frames for each video equally between the validation and test sets, taking every other frame from both *In-Action* and *Out-of-Action* frames. Each video contains evaluation frames spanning all difficulty tiers (easy, medium, hard). The size of the validation and test sets corresponds to only a fraction of the number of training frames due to strict constraints on the sampling of evaluation frames, which include high variability in viewpoints and a minimum time gap between the training and test/validation frames as described in Section 4.1 of the main paper.

For the *Hard (In-Action)* and *Medium (Out-of-Action)* settings, this time gap is set to 1s, which introduces increased difficulty for rendering novel views, since a significant portion of an activity might have taken place and neural rendering approaches would have to interpolate motion to account for this. While this is indeed a challenging task, it provides a unique opportunity for further explorations in neural rendering. We can account for the ambiguity that this choice introduces in two ways: resort to an evaluation protocol that accounts for that (e.g., best-of-K prediction) or accept that pixel predictions will have to be approximate for dynamic pixels and still measure the PSNR score. While the latter is not perfect, it is still reasonable for most 1s gaps and is much simpler than alternatives. The preference for this choice is also common in other ambiguous prediction tasks; for example, in the GTA-IM benchmark, where 3D path error is estimated after 0.5, 1, 1.5, and 2s [2],

Table 1: **EPIC Fields splits statistics**. We summarise the frame count and average frames per video for each split and for different difficulties (Easy, Medium, Hard). The number of frames for the validation and test sets is only a fraction of the training frames. This is due to strict constraints on the sampling of evaluation frames such as a high variety of viewpoints and the minimum time frame between train and test/validation frames. The train frames are fixed, regardless of the difficulty level.

| | In-Action (Easy) | | Out-of-Action (Medium) | | Out-of-Action (Hard) | | Total | |
|---|---|---|---|---|---|---|---|---|
| | #frames | average | #frames | average | #frames | average | #frames | average |
| Train | — | — | — | — | — | — | 103,571 | 2071.42 |
| Val | 3,448 | 68.96 | 657 | 13.14 | 305 | 6.1 | 4,410 | 88.2 |
| Test | 3,461 | 69.22 | 695 | 13.9 | 289 | 5.78 | 4,445 | 88.9 |

the TrajNet benchmark, where prediction is estimated for 4.8s from the observed frame [1], and the future hand prediction task in Ego4D, which uses a time gap of 1.5s from the observed frame [5].

For the *Easy (Out-of-Action)* setting, there is no temporal gap between training and evaluation frames and no specific action taking place. Consequently, both the complexity for rendering novel views and the ambiguity in evaluation are reduced for this subset of frames. This simplified setup parallels existing NVS benchmarks.

# 5 Additional training details for benchmarks

We now provide precise hyperparameters for the baselines used in the NVS and UDOS benchmarks. We provide full code for reproducing these results with the publication.

**NeRF-W, T-NeRF+, NeuralDiff.** We base our implementation of all 3D baselines on the codebase from NeuralDiff [10] and merge the other two approaches into the same PyTorch [9] codebase to align all training and evaluation details between models. We use the same training setup as in NeuralDiff, which involves training one model per baseline on each scene, taking approximately 12 hours using one NVIDIA Tesla P40 per experiment. Furthermore, the models are trained with hierarchical sampling (with a coarse and fine model as in the typical NeRF setting) and with a batch size of 1024. We train with the Adam optimizer for 10 epochs and an initial learning rate of $5 \times 10^{-4}$ that is adjusted during the training with a cosine annealing schedule.

**MG.** We use the provided code and train the model on our training split frames, jointly, for 135k iterations with a batch size of 32 and a learning rate of $5 \times 10^{-4}$.

**STM and XMEM.** For STM, we finetune a pretrained COCO [8] model on VISOR for 400K iterations with a batch size of 32 and a learning rate of $1 \times 10^{-5}$. For XMEM, we use the pretrained YoutubeVOS [11] model published in the XMEM paper and finetune it on VISOR for 100K iterations, with a batch size of 16 and a decaying learning rate initialised with $1 \times 10^{-5}$.

In the main paper, we include some qualitative results for the VOS challenge for a single object. We add more examples showing multi-object segmentation in Figure 7. The figure shows samples of failures of *Fixed 2D* in scenarios (a) and (b) and a case when both *Fixed 2D* and *Fixed 3D* fail to segment the dynamic objects (c).

# 6 EPIC Fields pipeline for Ego4D videos

While our reconstruction pipeline addresses several difficulties that are inherent to the videos of EPIC-KITCHENS [3], we can also apply it to other ego-centric videos such as the ones from Ego4D [6]. Using the pipeline as is, we can estimate camera poses for Ego4D videos that are about cooking and construction/building. We showcase this through an example in Figure 8 and two videos of reconstructions and camera tracks:

- Task: Construction —- 35 minutes of decorating and refurbishment. The video at `https://youtu.be/EZlayZIwNgQ` contains situations of challenging camera pose estimation including the camera wearer on a ladder (01:29, 05:07), kneeling down (16:14), as well as

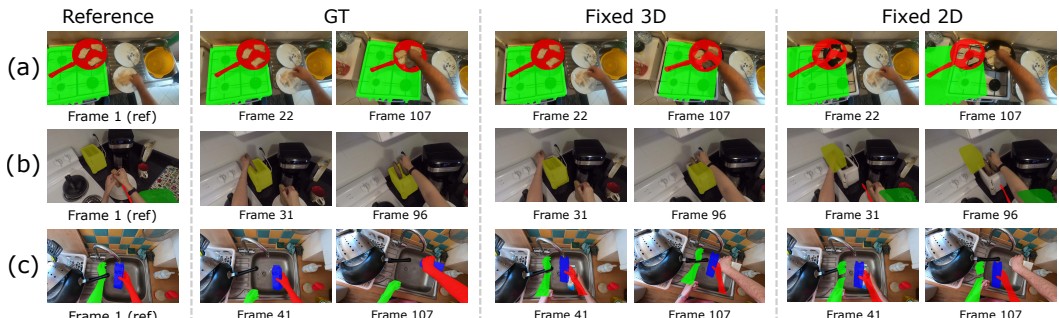

Figure 7: **Qualitative results for Semi-Supervised VOS.** We show samples with multiple objects. In scenarios (a) and (b), the *Fixed 3D* baseline effectively handles static objects, whereas the *Fixed 2D* falters due to camera movement. Conversely, in scenario (c), both strategies prove unsuccessful as the objects are in motion, invalidating the presumption of fixed objects in either 3D (*Fixed 3D*) or 2D (*Fixed 2D*), hence their failure.

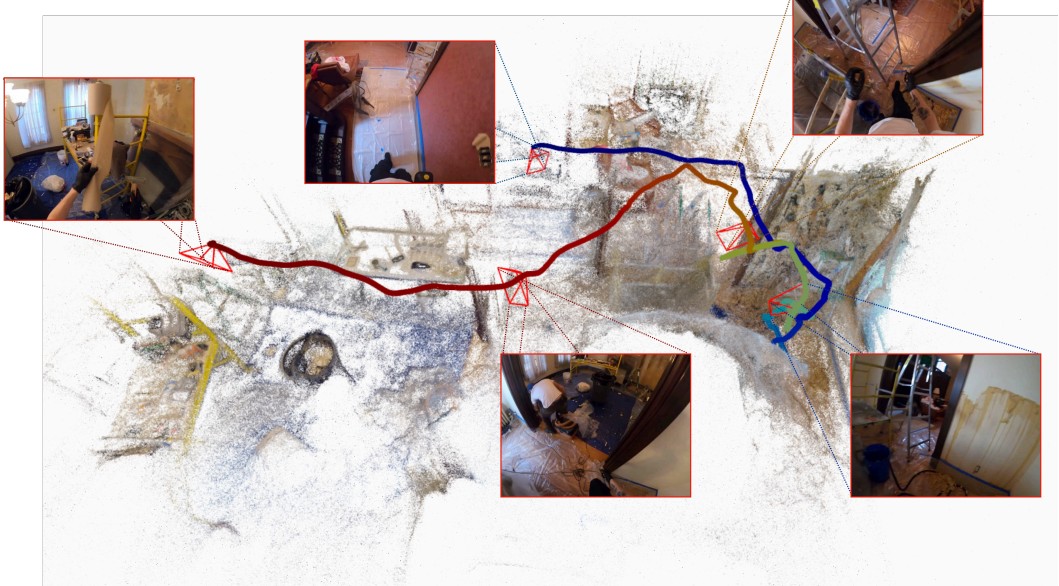

Figure 8: Visualisation of the 3D reconstruction for one video of the Ego4D dataset capturing building and refurbishment activities, with camera estimated using the EPIC Fields pipeline

drinking and navigating the scene (27:25) amongst many interesting poses. (Ego4D video a2dd8a8f-835f-4068-be78-99d38ad99625, source: CMU US)
- Task: Cooking —- 10 minutes. The corresponding video can be found at `https://youtu.be/GfBsLnZoFGs` (Ego4D video 18f5c2be-cb79-46fa-8ff1-e03b7e26c986).