# OpenReview forum: "EPIC Fields: Marrying 3D Geometry and Video Understanding"
_NeurIPS.cc/2023/Track/Datasets_and_Benchmarks — NeurIPS 2023 Datasets and Benchmarks Poster_

### Official Review · Reviewer_FW6P · 2023-07-18
**A nice work combining 3d geometry and video understanding**

**Rating:** 7
**Confidence:** 4
**Correctness:** The claims are correct.
**Clarity:** The paper is well written.

**Strengths:**

The authors focus on the significance of 3D information in video understanding datasets. A novel pre-processing step is applied to the classic Structure-from-Motion method in order to acquire reliable reconstruction quality.

In Section 4, the authors list three important tasks and show advantages of using 3D camera information in these tasks. These results, the proposed benchmark, and the codebase will provide valuable insights and an experimental playground to further investigate methods that combine geometry and video understanding.


**Additional Feedback:**

Nothing.

**Documentation:**

The dataset can be accessed at http://epic-kitchens.github.io/epic-fields.

**Ethics:**

Nothing.

**Limitations:**

The authors clearly addressed the societal impact of this work. better AR may be used for deception and many capabilities powering an assistant may also aid surveillance.

**Opportunities For Improvement:**

The authors claim that filtering the frames and addressing the skew is a key component to the success of reconstruction. However, neither experiments nor statistical analysis founded in Section 3 show that reconstruction performs better after filtering.

**Relation To Prior Work:**

EPIC Fields attaches 3D camera information to prior dataset EPIC-KITCHENS.

**Summary And Contributions:**

The authors of this paper mainly make two contributions.

Firstly, the dataset EPIC-KITCHENS is extended to EPIC Fields, which attaches camera information to most frames of the origin videos. A novel pre-processing step is designed to increase the reliability and speed of reconstruction.

Secondly, three benchmark tasks, which are novel view synthesis, dynamic objects identifying and video object segmentation, are proposed to demonstrate the advantages of 3D information.

---

> ### Author Response · Authors · 2023-08-19
> **Author response to reviewer FW6P**
>
> > **Opp4Imp:** The authors claim that filtering the frames and addressing the skew is a key component to the success of reconstruction. However, neither experiments nor statistical analysis founded in Section 3 show that reconstruction performs better after filtering.
>
> The impact of different frame sampling strategies is already assessed quantitatively in Table 1 in the sup. mat. We devote Sec 4.1 in supplementary to showcasing the impact of the sampling on reconstructions. As shown in Table 1 in supplementary, only 21/30 videos were reconstructed with uniform sampling - the rest indeed fail altogether. Importantly, even for the 21 videos, the reconstructions are both quantitatively and qualitatively unsatisfactory. This can be shown in the number of registered points and reconstruction error in Table 1 but also more importantly qualitatively.
>
> We also provide a qualitative comparison at this [link](https://drive.google.com/file/d/1pjytjQQj5XGbZ0V4uVNaoQjtYZ9EElFN/view?usp=sharing) that showcases the reconstructions of three videos with our proposed pipeline and with uniform sampling. It is clear that sampling is indeed a key component. For example, in page 1, just the corner of the kitchen was reconstructed by uniform sampling whereas our proposed pipeline reconstructed almost the entire kitchen. This is primarily due to the skewed distribution of viewpoints as noted by the reviewer. In other examples, e.g. in page 2, the reconstruction of the sink only is quite noisy.
>
> To clarify further:
>
> SfM methods weigh all input views equally and thus naturally “overfit” to certain camera viewpoints if they are over-represented. Common practice, as advised by the COLMAP authors is to [subsample video frames](https://github.com/colmap/colmap/issues/388#issuecomment-397589814) and to use a [larger diversity of scene viewpoint](https://github.com/colmap/colmap/issues/677#issuecomment-529187457).
>
> In principle, one could probably modify COLMAP to automatically weigh down redundant viewpoints. Nonetheless, our attempts to run COLMAP with the full set of frames from a video failed to achieve convergence within a _week_, which suggests that downsampling is necessary for scalability in any case.
>
> Hence, our approach to downsampling frames addresses both issues simultaneously.

---

### Official Review · Reviewer_cueE · 2023-07-21
**Applied COLMAP to EPIC-KITCHENS and made some benchmarks.**

**Rating:** 7
**Confidence:** 5
**Clarity:** Yes, very clear.

**Strengths:**

1) Providing the camera poses in the second large dataset in egocentric vision is indeed beneficial.
2) The three new benchmarks can be a test bed for the novel methods in the future.

**Additional Feedback:**

I'm happy to increase the scores if the authors address 4, 5, and 6 in *Opportunities For Improvement" and the minor point in "Correctness".  I do understand that directly evaluating the camera poses with missing ground truth or providing the camera poses on Ego4D is asking for too much.

**Correctness:**

See above "Opportunities For Improvement". 2 and 6 can be moved here?

A minor point.
> 127 dynamic objects. Other challenges include the video length (avg 9 mins) and the skewed distribution\
> 128 of viewpoints: videos have alternating phases of small motion around hot-spots (e.g., cooking at a\
> 129 hob or washing at the sink) and fast motion between hot-spots (e.g., moving the pot to the sink).

> 130 We address these challenges by: (1) intelligently filtering each video to reduce the number of frames\
> 131 and address the skew, a key component to our success;

> 135 Filtering. The goal of this step is to downsample a video while maintaining enough overlapping\
> 136 viewpoints for accurate reconstruction while reducing the viewpoint skew noted earlier.

It is not clear why this is considered "a key component to our success"! The paper should explain why having a skewed distribution of viewpoints is problematic in Structure-from-Motion (SfM). For readers who are not familiar with SfM (remember that half of the readers could be people in egocentric vision who are not well-versed in SfM), having many redundant viewpoints may not appear to be an issue. If there are more overlapping views with small camera motion, why not use them directly? Shouldn't SfM work better when the camera does not move much between frames? So, why did this paper choose to discard this valuable information by filtering and temporally downsampling it? In short, please clearly explain why having frames with substantial visual overlap is considered a problem.

**Documentation:**

Regarding the "Opportunities For Improvement" mentioned above, points 4 and 5 also pertain to documentation in a broader sense since they involve the release of code.

**Ethics:**

No ethical issue.

**Limitations:**

See above "Opportunities For Improvement". 1 and 3 can be moved here.

**Opportunities For Improvement:**

1) The camera poses are estimated by applying existing software (COLMAP) to the EPIC-KITCHENS and VISOR datasets, without any new manual annotations or data collection.
2) The direct evaluation of the accuracy of the estimated camera poses is missing. Since there are no ground truth camera poses, the only indirect evaluation provided is the reprojection error of the reconstructions. However, the paper does not discuss the potential effects of camera pose estimation errors on the proposed benchmarks.
3) There is no legitimate reason provided for why this paper exclusively focused on EPIC-KITCHENS. Ego4D is a larger and more diverse dataset these days, so the paper should also provide camera poses for Ego4D if possible.
4) Only the code to load and visualize the data is released, while the code for camera pose estimation is missing. In short, I want the code to fully reproduce the Sec 3.2. This prevents users from applying the same method to obtain camera poses in other egocentric vision datasets such as Ego4D.
5) The code to reproduce the baselines on the three benchmark tasks is also missing. This hinders the ability to replicate the benchmark baselines. Although it's likely that the authors used existing codebases of the baseline methods (e.g., NeRF), it would be better to provide the preprocessed files, etc., along with a link to the specific code. This way, other researchers can easily run the script and reproduce the baselines.
6) The baselines in semi-supervised video object segmentation are not sufficient. The "Fixed in 2D" and "Fixed in 3D" methods are very naive. The paper should also test state-of-the-art semi-supervised video object segmentation methods published in recent top conferences. While it's not the reviewer's responsibility to point out which SOTA method to use, the authors should make an effort to identify suitable SOTA methods with citations. This has been done for the first two benchmark tasks, but not for semi-supervised video object segmentation.

**Relation To Prior Work:**

Yes adequate.

**Summary And Contributions:**

This paper uses COLMAP to obtain camera parameters on the EPIC-KITCHENS dataset. The contributions of this work are as follows:

1) The release of camera poses for 90+% of the video frames in the EPIC-KITCHENS dataset.

2) The introduction of three benchmarks for novel view synthesis, unsupervised dynamic object segmentation, and semi-supervised video object segmentation, along with some baseline results.

---

> ### Author Response · Authors · 2023-08-19
> **Author response to reviewer cueE [1/2]**
>
> > **Opp4Imp1:** The camera poses are estimated by applying existing software (COLMAP) to the EPIC-KITCHENS and VISOR datasets, without any new manual annotations or data collection.
>
> If the question is about whether our contribution is significant to the community, we note that it required 9,056 GPU hours to estimate the cameras (the registration processes took 1,695 and 569 computation hours, on 4 GPUs),  without considering the work required to tune 3D reconstruction, ensuring quality control, and running the baseline experiments. Hence, obtaining this data is very laborious and shipping it to the community carries significant research value.
>
> > **Opp4Imp2:** The direct evaluation of the accuracy of the estimated camera poses is missing. Since there are no ground truth camera poses, the only indirect evaluation provided is the reprojection error of the reconstructions. However, the paper does not discuss the potential effects of camera pose estimation errors on the proposed benchmarks.
>
> As observed by the reviewer, there is no way to compute quantitatively the camera pose estimation error as there are no ground truth poses. Yet, we manually and individually checked each video by highlighting real-world (in 3D) lines and verifying that their projection based on the estimated pose is reasonable and stable in 2D. Sample images of 3D lines and corresponding videos are available [here](https://drive.google.com/drive/folders/16t9fqSfiXk3ac95jfrzvWN1yxU0xIHGR).
>
> Indicatively, we also note that the PSNR scores of our NeRF reconstructions are in line with the PSNR scores obtained by others on datasets like ScanNet and Matterport [Dense Depth Priors for Neural Radiance Fields from Sparse Input Views. Roessle et al., CVPR 2022] on static scenes, which suggests that our camera reconstruction quality is likely in the same ballpark as these other datasets.
>
> > **Opp4Imp3:** There is no legitimate reason provided for why this paper exclusively focused on EPIC-KITCHENS. Ego4D is a larger and more diverse dataset these days, so the paper should also provide camera poses for Ego4D if possible.
>
> The reason why we base our work on EPIC-KITCHENS is that our dataset EPIC Fields builds on the VISOR annotations of hands and action-relevant objects which form the ground truth for our benchmarks (L57). This is currently unavailable for other egocentric datasets including Ego4D.
>
> Ego4D contains a mix of videos - some suitable for our pipeline. This is because some videos are purely navigational outdoors, others include work in challenging settings (e.g., gardening) where we do not believe any pipeline, including ours, can accurately predict camera pose estimates. However, a subset of these Ego4D videos are suitable for being reconstructed via our proposed pipeline. There is no clear meta-data to know the exact subset.
>
> As we noted in the general comments and to address ZiM7’s concern and prove the suitability of our pipeline, we have also reconstructed camera poses from two videos of Ego4D (see general comments, [video1](https://youtu.be/EZlayZIwNgQ), [video2](https://youtu.be/GfBsLnZoFGs)). In addition, we have made [our pipeline public](https://github.com/epic-kitchens/epic-fields-code#reconstruction-pipeline), enabling the broader community to apply it to different videos as well.
>
> > **Opp4Imp4:** Only the code to load and visualise the data is released, while the code for camera pose estimation is missing. [...] This prevents users from applying the same method to obtain camera poses in other egocentric vision datasets such as Ego4D.
>
> As mentioned above and in the general comments, we have released the [code](https://github.com/epic-kitchens/epic-fields-code) for the camera pose estimation. The reviewer, as well as the whole research community, can fully reproduce Sec 3.2. We already showcase that our camera pose estimation pipeline can be used for other egocentric videos, such as those from Ego4D.
>
> > **Opp4Imp5:** The code to reproduce the baselines on the three benchmark tasks is also missing. [...] it would be better to provide the preprocessed files, etc., along with a link to the specific code. This way, other researchers can easily run the script and reproduce the baselines.
>
> Thanks to the reviewer for their request. As we stated in the checklist, we indeed want to release all code to replicate the benchmarks. As we are still in the process of cleaning this code to make it easier to use by others, but for now, we provide two repositories for the reviewer to check and confirm. These are available at:
> * For the New-View Synthesis (NVS, Task 1) and Unsupervised Dynamic Object Segmentation (UDOS, Task 2) we provide the evaluation [code](https://github.com/dichotomies/epic-fields-nvs-udos).
> * For the Semi-Supervised VOS we provide the [code](https://github.com/AhmadDarKhalil/epic-fields-vos) to replicate the baselines.
>
> The code and instructions will be further cleaned and shared with the camera-ready version of the paper.

---

> ### Author Response · Authors · 2023-08-19
> **Author response to reviewer cueE [2/2]**
>
> > **Opp4Imp6:** The baselines in semi-supervised video object segmentation are not sufficient. The "Fixed in 2D" and "Fixed in 3D" methods are very naive. The paper should also test state-of-the-art semi-supervised video object segmentation methods published in recent top conferences. While it's not the reviewer's responsibility to point out which SOTA method to use, the authors should make an effort to identify suitable SOTA methods with citations. [...]
>
> We thank the reviewer for the suggestion, we have now provided additional results for the semi-supervised VOS becnhmark with two methods: STM [A] and XMem [B]. Note that XMem is the state of the art in semi-supervised VOS on YouTube VOS 2018 [C] according to PapersWithCode. A full table of results is provided below:
>
> |Method|3D||Static|||SS+Dyn|||VISOR VAL||
> |-|-|:-:|:-:|:-:|:-:|:-:|:-:|:-:|:-:|:-:|
> |||J&F|J|F|J&F|J|F|J&F|J|F|
> |Fixedin2D|\-|17.8|23.8|11.6|12|11.9|12|12.5|13.4|11.6|
> |Fixed3D|✓|48.4|52.2|44.6|29.6|27.8|31.5|31.3|30.5|32.2|
> |Pretrained STM|\-|64.3|65.4|63.1|63.7|60.8|65.5|63|60.8|65.2|
> |Fine-tuned STM|\-|76.8|77.7|76|76.6|73.8|79.5|76.4|74.2|78.6|
> |Pretrained Xmem|\-|63.2|64|62.5|64.1|61.1|67.1|64|61.5|66.4|
> |Fine-tuned XMem|\-|77|77.7|77.4|78|75.3|80.7|77.3|75.2|79.4|
>
> As shown in the table, learning-based methods significantly improve performance in both the Static and SS+Dyn scenarios. Their effectiveness comes from their training approach which uses image features to segment objects, rather than just naive tracking. However, it is important to note that both STM and XMem primarily utilize 2D information. There remains an untapped opportunity in harnessing the benefits of integrating 3D data for addressing this task.
>
> [A] Oh et al., Video Object Segmentation using Space-Time Memory Networks. ICCV 2019
>
> [B] Cheng et al., XMem: Long-Term Video Object Segmentation with an Atkinson-Shiffrin Memory Model. ECCV 2022
>
> [C] Xu et al. Youtube-vos: A large-scale video object segmentation benchmark. arXiv preprint arXiv:1809.03327
>
> > **Minor point:** It is not clear why this is considered "a key component to our success"! The paper should explain why having a skewed distribution of viewpoints is problematic in Structure-from-Motion (SfM). [...] If there are more overlapping views with small camera motion, why not use them directly? Shouldn't SfM work better when the camera does not move much between frames? So, why did this paper choose to discard this valuable information by filtering and temporally downsampling it? In short, please clearly explain why having frames with substantial visual overlap is considered a problem.
>
> SfM methods weigh all input views equally and thus naturally “overfit” to certain camera viewpoints if they are over-represented. Common practice, as advised by the COLMAP authors, is to [subsample video frames](https://github.com/colmap/colmap/issues/388#issuecomment-397589814) and to use a [larger diversity of scene viewpoint](https://github.com/colmap/colmap/issues/677#issuecomment-529187457).
>
> In principle, one could probably modify COLMAP to automatically weigh down redundant viewpoints. Nonetheless, our attempts to run COLMAP with the full set of frames from a video failed to achieve convergence within a _week_, which suggests that downsampling is necessary for scalability in any case.
>
> Hence, our approach to downsampling frames addresses both issues simultaneously.
>
> Note that the impact of different frame sampling strategies is already assessed quantitatively in Table 1 in the sup. mat. We devote Sec 4.1 in supplementary to showcasing the impact of the sampling on reconstructions. As shown in Table 1 in the supplementary, only 21/30 videos were reconstructed with uniform sampling - the rest indeed fail altogether. Importantly, even for the 21 videos, the reconstructions are both quantitatively and qualitatively unsatisfactory. This can be shown in the number of registered points and reconstruction error in Table 1 but also more importantly qualitatively.
>
> We also provide a qualitative comparison at this [link](https://drive.google.com/file/d/1pjytjQQj5XGbZ0V4uVNaoQjtYZ9EElFN/view?usp=sharing) that showcases the reconstructions of three videos with our proposed pipeline and with uniform sampling. It is clear that sampling is indeed a key component. For example, on page 1, just the corner of the kitchen was reconstructed by uniform sampling whereas our proposed pipeline reconstructed almost the entire kitchen. This is primarily due to the skewed distribution of viewpoints as noted by the reviewer. In other examples, e.g. on page 2, the reconstruction of the sink only is quite noisy.

---

> ### Comment · Reviewer_cueE · 2023-08-19
>
> Thanks for the responses. The authors addressed my concerns, so I increased the score.

---

### Official Review · Reviewer_Pv5q · 2023-07-22
**Simple technique, useful extension to EPIC Kitchens**

**Rating:** 6
**Confidence:** 4
**Correctness:** yes
**Clarity:** yes

**Strengths:**

- Simple yet effective technique to solve for camera poses in egocentric long duration videos.
- The outcome (the augmented EPIC Kitchen dataset) is useful for the community to study dynamic 3D scene understanding.
- Interesting to see that in the UDOS benchmarking results, 3D methods indeed do better in semi static movable object segmentation.

**Additional Feedback:**

Overall I think the contribution is solid, but the dynamic view synthesis setup does sound weird to me. I'm happy to increase my score if this concern can be properly addressed or I get convinced by a rebuttal.

**Documentation:**

yes

**Limitations:**

yes

**Opportunities For Improvement:**

- The dynamic view synthesis experiment setup looks a bit weird to me. Why do you discard training frames that were within 1 second of a test frame in the in-action partition? This makes the task so difficult that I feel it is impossible to address with existing dynamic reconstruction methods. This means you have to do both view synthesis and motion interpolation in the interval of >1s. A lot can happen within one second when you are working in a kitchen, so the uncertainty may overwhelm, and then the pixel-level synthesis does not make too much sense -- the best you can do is to guess what might happen, but synthesizing accurately in pixel level does not sound possible to me.

**Relation To Prior Work:**

yes

**Summary And Contributions:**

This paper presents an augmentation to the EPIC Kitchen dataset by providing camera poses for it. This allows evaluating egocentric dynamic reconstruction tasks on EPIC Kitchen. This paper includes benchmark evaluation for dynamic novel view synthesis, Unsupervised Dynamic Object Segmentation (UDOS), and semi-supervised video object segmentation.

---

> ### Author Response · Authors · 2023-08-19
> **Author response to reviewer Pv5q**
>
> > **Opp4Imp:** The dynamic view synthesis experiment setup looks a bit weird to me. Why do you discard training frames that were within 1 second of a test frame in the in-action partition? This makes the task so difficult that I feel it is impossible to address with existing dynamic reconstruction methods. This means you have to do both view synthesis and motion interpolation in the interval of >1s. A lot can happen within one second when you are working in a kitchen, so the uncertainty may overwhelm, and then the pixel-level synthesis does not make too much sense -- the best you can do is to guess what might happen, but synthesising accurately in pixel level does not sound possible to me.
>
> First, this is indeed a challenging task, but it is good for a new dataset not to be oversaturated too quickly. For this reason, we have devised a set of NVS tasks of varying degrees of difficulty. We note that our pipeline allows creating more sets of train/val/test splits with bigger/smaller gaps between the frames. We also note that the PSNR scores for the Medium and Hard-BG settings are relatively close to those of the Easy one (i.e., the setting that is the closest to current NVS benchmarks).
>
> We opt for selecting a gap of 1s which is indeed more ambiguous in the Hard (in-action) setting. Here, there are two choices: resort to an evaluation protocol that accounts for this ambiguity (e.g., best-of-K prediction) or accept that pixel predictions will have to be approximate for dynamic pixels and still measure PSNR. While the latter is not perfect, it is still reasonable for most 1s gaps and is much simpler than alternatives. We also note similar ambiguous prediction tasks in computer vision that are benchmarked in a similar spirit. Just to provide a few examples of unimodal prediction metrics:
>
> * The GTA-IM benchmark where 3D path error is estimated after 0.5, 1, 1.5 and 2 seconds  [Long-term human motion prediction with scene context. Cao, Gao, Mangalam, Cai, Vo, Malik. Proc. ECCV, 2020]
> * The TrajNet benchmark where prediction is estimated for 4.8 seconds from the observed frame [An evaluation of trajectory prediction approaches and notes on the TrajNet benchmark. Becker, Hug, Hübner, Arens. arXiv.cs, abs/1805.07663, 2018.]
> * Future hand prediction is estimated after 1.5 seconds from the observed frame [Ego4D: Around the World in 3,000 Hours of Egocentric Video, CVPR22]
>
> In fact, our task is less ambiguous than these as we only need to interpolate between seen past and future observations for rather short time spans.

---

### Official Review · Reviewer_ZiM7 · 2023-07-25
**Great dataset including additional information has great potential**

**Rating:** 7
**Confidence:** 4

**Strengths:**

1) This paper is very helpful for the whole research community as it provides the camera poses + some other additional information for the kitchen-visor dataset which lift large-scale 2D annotations to 3D. This would be essential to study of 3D egocentric trajectories concerning actions (for anticipation), objects (for understanding object state changes), and the interactions between hands and objects.
2) The proposed three benchmarks (dynamic new-view synthesis, unsupervised dynamic object segmentation, video object segmentation), and the related codebase are valuable for further exploration of 3D and video understanding.
3) The paper highlights several technical innovations that were crucial in constructing the EPIC Fields dataset. These innovations likely address challenges in processing egocentric videos, ensuring that the resulting dataset is of high quality and reliable for researchers to use.  It also successfully processes a substantial amount of data, including 671 videos spanning 45 different kitchens, resulting in over 18 million registered video frames with estimated camera poses



**Additional Feedback:**

None

**Clarity:**

Overall the paper states the dataset construction process and benchmark clearly. The paper is well structured: the abstract provides a good review and introduction motivates the research project. But it's better to polish the paper writing and the sentences.

**Correctness:**

The dataset construction process makes sense. By filtering the video, using SfM to reconstruct and registering remaining frames to this reconstruction, it solves the challenges of dynamic scene. I am wondering did you spend human labor to make sure the dataset quality is high after those process?

The benchmark and experiment design is performed correctly and would provides a good example for future research.

**Documentation:**

I think there is sufficient detail on availability and maintenance in the paper. The authors provided the URL and release the dataset and the website/dataset is well maintained. There is also sufficient detail on ethical use.

**Ethics:**

I don’t suspect there are ethical concerns with the submission. Participants recorded themselves and approved their footage. They also De-ID to avoid potential privacy issues.

**Limitations:**

Please see the "Opportunities For Improvement". It would be great if you could explain more comparing with Ego4D

**Opportunities For Improvement:**

1) What is the advantages of this dataset comparing with Ego4D?
2) Based on line 233 to 242 in Section 4.2, 3D based methods are better suited for discovering semi-static objects that are not currently in motion, 2D-based methods are good at discovering dynamic objects. In Table 3, NeuralDiff [54] is a 3D-based method, but it performs much better on Dynamic objects than the other two 3D-based methods NeRF-W [30] and T-NeRF+ [13]. Could you explain the reason for that?

**Relation To Prior Work:**

It already compared with some datasets in this field but it would be better to see more discussion on Ego4D and H2O.

**Summary And Contributions:**

This paper discusses the integration of 3D visual geometry and image understanding. Recent advancements in neural rendering have allowed for a closer integration of machine learning in tasks like reconstruction and rendering, creating an opportunity to combine 3D geometry and video understanding. The authors propose extending the EPIC-KITCHENS action understanding dataset with visual geometry to create EPIC Fields. This new dataset includes reconstructed 3D cameras and benchmark tasks rooted in 3D geometry and semantic visual understanding.

The contributions of the paper are:
1) Augmentation of EPIC-KITCHENS: The authors extend the EPIC-KITCHENS dataset with camera information, addressing the lack of suitable development data for dynamic content. By providing camera information for egocentric videos, the authors enable the study of geometric reconstruction and semantic understanding in long and highly dynamic videos.
2) Benchmark Tasks: With the EPIC Fields dataset, the authors introduce benchmark tasks such as dynamic novel view synthesis, unsupervised dynamic object segmentation, and video object segmentation. These tasks are designed to challenge existing approaches and pave the way for researchers to investigate new questions related to 3D understanding.

---

> ### Author Response · Authors · 2023-08-19
> **Author response to reviewer ZiM7**
>
> > **Opp4Imp1:**  Advantages of this dataset comparing to Ego4D
>
> We have based our dataset and benchmarks on EPIC KITCHENS. The EPIC KITCHENS dataset offers two significant advantages which justify our choice.
>
> First, it offers multiple lengthened recordings in the same environment - participants record all their kitchen activities for 3 consecutive days. Such recordings offer the ability for detailed exploration of the full 3D space.
>
> Second, EPIC Fields builds on the VISOR annotations of hands and action-relevant objects which form the ground-truth for our benchmarks (L57). This is particularly helpful for the dynamic view synthesis benchmark where we can use the VISOR annotations as ground truth of dynamic objects. Such annotations are currently unavailable for Ego4D videos.
>
> We note, however, that our pipeline, now [publicly available](https://github.com/epic-kitchens/epic-fields-code), can also be used on Ego4D videos of indoor locations. A large proportion of Ego4D videos capture outdoor navigation and workers seated in one location (e.g., table) carrying out a task (e.g., knitting). These videos are not suitable for reconstructing the 3D scene. However, a subset of Ego4D videos does indeed capture camera wearers carrying out cooking, cleaning or maintenance tasks. Unfortunately, there is no meta-data available to identify these videos specifically from the 100s of videos in the massive-scale Ego4D dataset.
>
> As noted in the general comments, we have applied our pipeline on 2 long videos ([video1](https://youtu.be/EZlayZIwNgQ),[video2](https://youtu.be/GfBsLnZoFGs)) from Ego4D and showcase the camera pose estimates, similar to our visualisation on EPIC Fields. Note that we cannot use these videos for our benchmarks, as there are no segmentation ground truths of moving objects, but our pipeline can indeed be used to provide camera pose estimates for a variety of tasks.
>
> > **Opp4Imp2:**  Based on line 233 to 242 in Section 4.2, 3D based methods are better suited for discovering semi-static objects that are not currently in motion, 2D-based methods are good at discovering dynamic objects. In Table 3, NeuralDiff [54] is a 3D-based method, but it performs much better on Dynamic objects than the other two 3D-based methods NeRF-W [30] and T-NeRF+ [13]. Could you explain the reason for that?
>
>
> Current 3D methods, including neural radiance fields, handle static parts of a scene better (including semi-static objects "not currently in motion") but struggle with dynamic objects. As noted by the reviewer, the latter is not true for NeuralDiff [54]. NeuralDiff has been specifically designed to tackle the task of segmenting moving objects (detachable or semi-static objects and dynamic objects), which explains its good performance compared to the other 3D-based methods.
>
> > **Relation to prior work:** It already compared with some datasets in this field but it would be better to see more discussion on Ego4D and H2O.
> > **Clarity:** more discussion on Ego4D and H2O
>
> We note the distinction between this dataset and Ego4D in Opp4Imp1 above. We will include this discussion in the paper as well as reconstructions on a selected number of videos.
> H2O (Kwon et al, ICCV 2021) captures a single person seated on a table performing table-top activities. All the data is captured on the same table - there is no human movement involved and no scene to reconstruct, apart from the single table. It is not clear to us how this dataset, extremely useful for object pose estimation, is relevant to our attempt to recover camera poses from complex egocentric videos. Additionally, as the viewpoints of H2O remain restricted to the table, it’s not ideal for evaluating novel view synthesis tasks.
>
> > **Correctness:** Did you spend human labor to make sure the dataset quality is high after those process?
>
> Yes. we did manually verify the estimated camera poses for each video before release. As ground-truth camera poses are unavailable, we performed the following thorough visual inspections:
> * We created and inspected the point cloud obtained from the reconstruction of each video, looking for any visual evidence of an error, such as duplicated elements, a surface appearing twice, etc.
> * We also manually annotated semantically meaningful 3D lines and used the estimated camera poses to project those in each 2D frame of a video. We verified that the 2D projections of the 3D lines were correct and consistent across the video. In practice, this simple test is quite sensitive and can easily spot bad cameras. Examples of such rendered images and videos are available from [google-drive](https://drive.google.com/drive/folders/16t9fqSfiXk3ac95jfrzvWN1yxU0xIHGR). For each example you can find the image of the 3D line and a corresponding 2D video of its projection. Note that our lines extend indefinitely just for simplicity but they typically correspond to an edge in the kitchen at parts.

---

### Author Response · Authors · 2023-08-19
**Global response to all reviewers**

We thank the reviewers for carefully reading our submission and providing constructive and insightful feedback. We are glad that all four reviewers found our dataset to be “helpful”, “useful”, “beneficial” and “insightful” to the research community. ZiM7 notes that we bring several technical innovations, Pv5q notes our techniques to be effective. FW6P notes our three tasks are important and cueE believes our tasks would provide a testbed for novel future methods.

We note a few things that are relevant to all reviewers:

1. We have already released our pipeline codebase for public use, available from: https://github.com/epic-kitchens/epic-Fields-code
This includes a step-by-step guideline and scripts to replicate the full pipeline to reconstruct camera poses.

2. We have verified our pipeline is applicable to other egocentric datasets. Using the pipeline as is, we can estimate camera poses for Ego4D videos of cooking and construction/building. We showcase this through 2 videos that we make available online for the reviewers to examine:

   * [Ego4D Video - Sample 1 - Construction](https://youtu.be/EZlayZIwNgQ) – 35 mins of decorating and refurbishment. The video contains situations of challenging camera pose estimation including the camera wearer on a ladder (01:29, 05:07), kneeling down (16:14), as well as drinking and navigating the scene (27:25) amongst many interesting poses. (Ego4D video a2dd8a8f-835f-4068-be78-99d38ad99625, source: CMU US)
   * [Ego4D Video - Sample 2 - Cooking](https://youtu.be/GfBsLnZoFGs) – 10 mins (Ego4D video 18f5c2be-cb79-46fa-8ff1-e03b7e26c986)

3. We have made available two repositories that replicate the baselines used in our paper.
* For the New-View Synthesis (NVS, Task 1) and Unsupervised Dynamic Object Segmentation (UDOS, Task 2) we provide code [here](https://github.com/dichotomies/epic-fields-nvs-udos).
* For the Semi-Supervised VOS, code to replicate the two baselines is available [here](https://github.com/AhmadDarKhalil/epic-fields-vos).

We are still cleaning these repositories for optimal use and will add more details to the README as well as the code for training before the camera ready deadline. In the meanwhile, the reviewers can replicate all baselines by accessing the links above.

We also provide detailed responses to each reviewer below.

Based on their suggestions, the main change we foresee in our manuscript is adding a discussion on the datasets and further qualitative examples of the frame sampling – these are included in the responses below. We will be providing these in the revised version of the paper.

---

### Decision · Program_Chairs · 2023-09-22

**Decision:**

Accept (Poster)

**Comment:**

All four reviewers liked the dataset and the proposed tasks and recommended acceptance.  The AC agrees.  The authors are encouraged to revise the submission based on the reviews in the camera-ready version.